# Tiny models from tiny data: Textual and null-text inversion for few-shot distillation

## Abstract

Few-shot learning deals with problems such as image classification using very few training examples. Recent vision foundation models show excellent few-shot transfer abilities, but are large and slow at inference. Using knowledge distillation, the capabilities of high-performing but slow models can be transferred to tiny, efficient models. However, common distillation methods require a large set of unlabeled data, which is not available in the few-shot setting. To overcome this lack of data, there has been a recent interest in using synthetic data. We expand on this line of research by presenting a novel diffusion model inversion technique (TINT) combining the diversity of textual inversion with the specificity of null-text inversion. Using this method in a few-shot distillation pipeline leads to state-of-the-art accuracy among small student models on popular benchmarks, while being significantly faster than prior work. Popular few-shot benchmarks involve evaluation over a large number of episodes, which is computationally cumbersome for methods involving synthetic data generation. We also present a theoretical analysis on how the accuracy estimator variance depends on the number of episodes and query examples, and use these results to lower the computational effort required for method evaluation. Finally, to further motivate the use of generative models in few-shot distillation, we demonstrate that our method outperforms training on real data mined from the dataset used in the original diffusion model training. Source code is available at TBD [1].

## 1 Introduction

In recent years, deep learning has enabled unprecedented performance in most computer vision tasks. To reach state-of-the-art, increasingly large foundation models (Dosovitskiy et al., 2020; Touvron et al., 2021; Fang et al., 2023; Kirillov et al., 2023; Awais et al., 2023) are often used, trained on increasingly huge datasets (Schuhmann et al., 2022; Zhai et al., 2022). In contrast, many computer vision applications have the opposite requirements. Representative training data can be scarce due to privacy concerns (e.g. medical data) or other practical issues around data collection (e.g. industrial fault detection requiring destructive processes for obtaining training data). Lightning-fast inference is often desired in both embedded systems and to get good economy in cloud-hosted solutions.

Few-shot learning (Song et al., 2023) deals with learning in very label-constrained situations, typically down to 1-5 labelled examples per class. Classical few-shot methods study architectures that can quickly adapt to new tasks by feeding a new set of labelled *support examples* into the model. Common few-shot benchmarks such as miniImageNet (Vinyals et al., 2016) have traditionally been strictly interpreted in the sense that only training data from within the dataset itself may be used, but models trained on extra data are becoming more and more common (Samuel et al., 2024a; Trabucco et al., 2024). Hu et al. (2022) argue that the strict (no pretraining) interpretation of few-shot learning may lead to an undesirable divergence between the few-shot learning and semi-supervised learning (Van Engelen & Hoos, 2020) communities. We adhere to this view.

---

[1] released with the camera-ready version

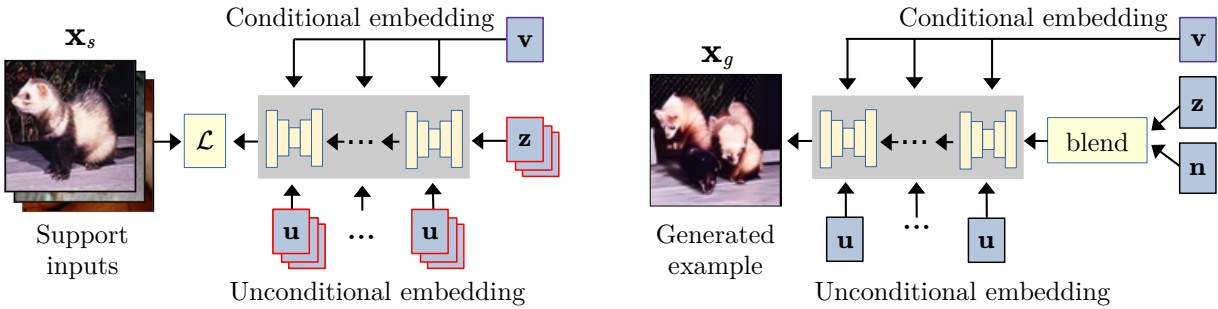

Figure 1: Overview of the TINT method. Left: From a set of input examples, we optimize all external quantities (latents and conditional/unconditional embeddings). Right: A new image is generated by blending a latent with noise and feeding it through the diffusion model.

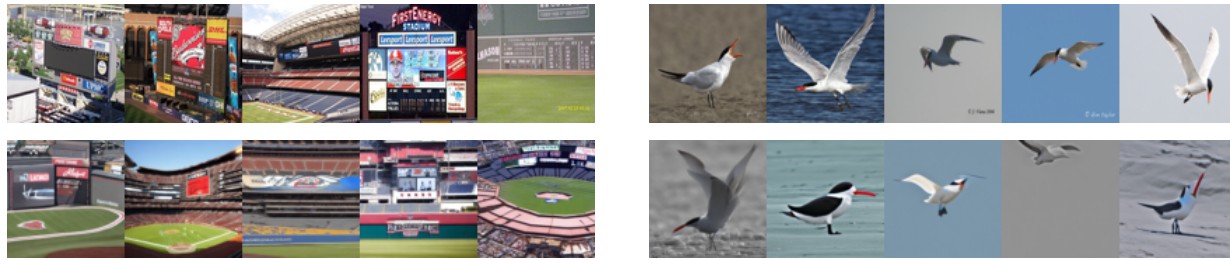

Figure 2: Examples of images generated by our method (bottom) from randomly selected support images (top). Left: Class *Scoreboard* from miniImageNet. Right: Class *Caspian Tern* from CUB.

In many practical applications, longer training times for adaptation to a specific task and the use of pre-trained models are often both acceptable, as long as the *final* model for a specific task has excellent performance in terms of speed and accuracy. To address this setting, we formulate our research question as: How can we maximize the accuracy of *tiny* efficient classifiers when only a *tiny* amount of application-specific labeled data is available, but when there are no restrictions on using common pre-trained models in the training pipeline, and a few GPU-hours of training for a specific task is acceptable? We emphasize that this is not an artificial scenario - rather, it is motivated from personal experience from MVP development of new industrial applications.

Large foundation models often present good few-shot transfer abilities when equipped with a small classifier head (Hu et al., 2022; Caron et al., 2021; Zhang et al., 2022). While such models may be required to handle *any* few-shot task well, a *specific* task (containing just a few classes) is likely solvable using a significantly smaller model. Using *knowledge distillation* (Hinton et al., 2015; Gou et al., 2021), a small, efficient classifier can be trained to mimic a large general few-shot model on a specific task. However, while typical distillation methods require a large set of representative unlabeled data (Hinton et al., 2015; Tian et al., 2019; Ahn et al., 2019), we only assume access to a few application-specific examples. This leads us to our overall approach: Adapt a generative model to produce a large set of images representative of the novel classes. Then use this data to distill knowledge from a well-performing (but large and slow) few-shot classifier to a small model, specialized on a single task. We base the generator on diffusion models, due to their impressive image generation capabilities (Rombach et al., 2022; Saharia et al., 2022; Ramesh et al., 2022; Balaji et al., 2022). A core challenge then becomes how to best specialize such models to produce data similar to the few-shot support examples. To this end, we propose a novel combination of textual inversion and null-text inversion, illustrated in Figure 1, and with some examples of generated training images shown in Figure 2.

Evaluation on traditional few-shot datasets (e.g. miniImageNet) is often done over a large number of randomly drawn episodes. This poses a practical problem for methods where specialization for each episode requires significant computational effort. In Section 3.3, we analyze the impact of the number of episodes and number of query examples per episode, and arrive at an evaluation setup reaching similar statistical significance with less compute. This makes it more practical to evaluate methods with heavy specialization procedures on traditional few-shot datasets.

Our main contributions can be summarized as:

- A novel diffusion inversion method (TINT), combining the diversity of textual inversion with the specificity of null-text inversion and outperforming baseline generation methods in the few-shot distillation context.

- Demonstrating that it is possible to significantly push the achievable accuracy of even tiny models such as the 113k-parameter *Conv4* by using them in a generative distillation framework.

- A theoretical analysis of the accuracy estimator variance in episode-based few-shot benchmarks, arriving at an evaluation procedure that is more practical for specialization-heavy methods.

- Demonstrating that using a generative model in few-shot distillation outperforms a direct use of the LAION dataset (Schuhmann et al., 2022) which was used to train the generative model in the first place.

## 2 Background

In this section, we briefly review the most important theory and prior work that our pipeline builds upon. This also serves for introducing our notation.

### 2.1 Few-shot classification

In *few-shot classification* (Vinyals et al., 2016; Ravi & Larochelle, 2016), the task is to classify a *query* $\mathbf{x}^{(q)}$ into one of $N$ classes identified by integer labels $y \in \{1, ..., N\}$, where each class is defined using a small set of labeled *support examples*. To simplify the notation, we focus on the case with $K$ support examples per class (*N-way, K-shot*), and use $\mathbf{x}_{n,k}^{(s)}$ to denote the $k$'th support example of class $n$. All theory transfers naturally to the case where $K$ is different for each class.

### 2.2 Diffusion Models

*Denoising diffusion probabilistic models* (Ho et al., 2020) model the distribution of a random variable $\mathbf{x}_0$ by transforming it into a tractable distribution (noise) over timesteps $t \in \{0, ..., T\}$. Our generator is built upon the popular *Stable Diffusion* (SD) (Rombach et al., 2022) model due to its public availability and use in prior work on diffusion inversion (Gal et al., 2022; Mokady et al., 2023). SD defines the diffusion process on a latent variable $\mathbf{z}_t$ of lower dimensionality than $\mathbf{x}_t$. The model is trained to minimize the loss

$$L = \mathbb{E}_{\mathbf{z}_0, t, \epsilon}\left[\|\epsilon - \epsilon(\mathbf{z}_t; t, \mathbf{v})\|^2\right], \tag{1}$$

where the noise estimation $\epsilon(\mathbf{z}_t; t, \mathbf{v})$ is implemented using a U-net (Ronneberger et al., 2015), and $\mathbf{v}$ is a CLIP embedding (Radford et al., 2021) of a text prompt fed as an additional input to this U-net. A VQ-VAE model (Razavi et al., 2019) maps between $\mathbf{x}_0$ and $\mathbf{z}_0$.

Using *Denoising Diffusion Implicit Models* (DDIM) (Song et al., 2020), the forward and backward processes can be made deterministic. We write this as $\mathbf{z}_{t+1} = f_t(\mathbf{z}_t)$, $\mathbf{z}_{t-1} = g_t(\mathbf{z}_t)$. We are mostly interested in the mapping between $\mathbf{z}_0$ and $\mathbf{z}_T$, writing it as $\mathbf{z}_T = F(\mathbf{z}_0)$ with $F = f_{T-1} \circ ... \circ f_0$ and $\mathbf{z}_0 = G(\mathbf{z}_T)$ with $G = g_1 \circ ... \circ g_T$. The strength of the conditioning can be controlled by *classifier-free guidance* (CFG)

(Ho & Salimans, 2022). For each timestep $t$, the noise is predicted using both a text embedding $\mathbf{v}$ and an unconditional embedding $\mathbf{u}$, and the noise prediction is shifted in the text-conditional direction according to

$$\epsilon_t = \epsilon(\mathbf{z}_t; t, \mathbf{u}) + \beta\big(\epsilon(\mathbf{z}_t; t, \mathbf{v}) - \epsilon(\mathbf{z}_t; t, \mathbf{u})\big). \tag{2}$$

When using classifier-free guidance, the DDIM inversion no longer reproduces the original input unless additional techniques are used. *Null-text inversion* (NTI) (Mokady et al., 2023) is one such technique. In NTI, latents $\mathbf{z}_T = F(\mathbf{z}_0)$ are first computed using DDIM. The original objective (Eq. 1) is then minimized with the unconditional embeddings $\mathbf{u}_{1:T}$ as variables (introducing a separate embedding $\mathbf{u}_t$ for each timestep $t$). The output of the inversion is the final latent $\mathbf{z}_T$ and adjusted embeddings $\mathbf{u}_{1:T}$.

*Textual inversion* (TI) (Gal et al., 2022) is a technique for making diffusion models output images that look similar (but not identical) to a small set of input images. The idea is to adjust the text embeddings of one or more words in a conditioning text prompt, essentially creating new words representing the input images. The optimization objective is still the original Eq. 1, but the variables to optimize are now selected embeddings of individual words (parts of $\mathbf{v}$).

## 2.3 Knowledge distillation

Knowledge distillation (Buciluă et al., 2006; Hinton et al., 2015; Gou et al., 2021) refers to training a *student* network to mimic a *teacher* network for the purpose of model compression or performance improvement. We base our pipeline on the *LabelHalluc* method Jian & Torresani (2022), where distillation is used in few-shot classification, using base classes as distillation training data, pseudo-labelled by a teacher sharing the same architecture as the student.

# 3 Method

We first describe our new diffusion inversion technique (Section 3.1), and then show how to apply it in a complete few-shot distillation pipeline (Section 3.2).

## 3.1 TINT

As distillation data, we would like new examples that are *similar* to the support examples, but still provide *sufficient variation* to make the student generalize to unseen query examples. This could be done by running null-text inversion (NTI) on the support examples, perturbing the latents and feeding them back through the diffusion model. However, the generated images would depart significantly from the right class if the level of perturbation is high, unless some guiding conditional input is applied. Using textual inversion (TI), we can construct such a conditioning input from the support examples. This leads to a combined method that we call TINT (Textual Inversion + Null-Text).

The method is illustrated in Figure 1 and works as follows: First, all support examples $\mathbf{x}_{n,k}^{(s)}$ of class $n$ are fed jointly to a TI procedure to find a text embedding $\mathbf{v}_n$ capable of generating similar examples. We use the generic prompt *a photo of an #object*, where *#object* represents the embedding to optimize. The embedding is initialized from the word *object*, i.e. we don't require a known text description of each support class. We then run NTI on each support example separately, conditioned by the embeddings obtained from TI. This produces latent vectors $\mathbf{z}_{T,n,k}$ at time $t = T$ and adjusted unconditional embeddings $\mathbf{u}_{1:T,n,k}$. Note that we have now optimized all embeddings present in the CFG procedure (Eq. 2) - both the conditional and unconditional ones.

Given the optimized embeddings, we want to make a random perturbation of a latent $\mathbf{z}_{T,n,k}$ and feed it back through the generative model $G$, using the adjusted $\mathbf{u}_{1:T,n,k}$ in the CFG. A naive way to do the random perturbation would be to blend $\mathbf{z}_{T,n,k}$ with noise, i.e. letting

$$\mathbf{z}' = (1 - \alpha)\mathbf{z}_{T,n,k} + \alpha\mathbf{n}, \quad \mathbf{n} \sim \mathcal{N}(0, \mathbf{I}), \quad \alpha \in [0, 1]. \tag{3}$$

However, as noted by Samuel et al. (2024a), a simple linear interpolation between two latents will often result in an output with lower norm than the inputs and lead to an overly bland, low-contrast image. To avoid this

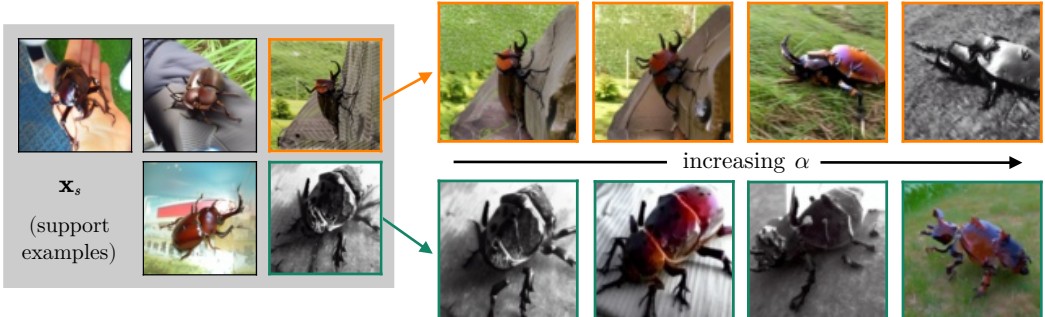

Figure 3: Generated images with increasing $\alpha$ (latent space noise), showing how the method gradually transitions from a more augmentation-like behavior to full synthetic image generation as $\alpha$ increases.

effect, we could simply rescale the latent norm by letting $\tilde{\mathbf{z}} = \mathbf{z}'\|\mathbf{z}'\|^{-1}$, but that would not reproduce $\mathbf{z}_{T,n,k}$ as $\alpha \to 0$. One option could be to use the norm-guided interpolation suggested by Samuel et al. (2024a), but we opt for a simpler approach. We compute a desired latent norm by interpolating the norms of the inputs to the blending operation from Eq. 3 and rescaling $\mathbf{z}'$ to this norm value, by

$$\tilde{\mathbf{z}} = \mathbf{z}'\|\mathbf{z}'\|^{-1}\big((1-\alpha)\|\mathbf{z}_{T,n,k}\| + \alpha\|\mathbf{n}\|\big). \tag{4}$$

To generate a new example, simply select a $k$ at random, draw a noise vector $\mathbf{n}$, interpolate and normalize using Eq. 3 and Eq. 4, and feed $\tilde{\mathbf{z}}$ into the diffusion model. Pseudo-code for the entire procedure is provided in Appendix A.1. As illustrated in Figure 3, when $\alpha$ is increased, our method gradually transforms from performing deep data augmentation (staying relatively faithful to the support examples) to complete synthetic data generation. For the few-shot case, we expect that a large $\alpha$ will work best, in order to provide as much variation as possible, while smaller $\alpha$ may be useful to in situations where a substantial amount of training data already exists.

Note that there is a significant resolution mismatch between available pre-trained Stable Diffusion models and popular few-shot classification datasets. We found that a naive upsampling before the textual inversion made the inversion prone to reproduce sampling artifacts, sometimes prioritizing this over semantically meaningful details. Using a pre-trained super-resolution (SR) model (Chen et al., 2023) for upscaling largely eliminated this problem. We also found it beneficial to apply the TI optimization target (Eq. 1) only for a restricted range $t \in [250, 1000]$, as including smaller $t$ made the textual inversion more prone to focus on remaining sampling artifacts. See Appendix B.2 for more details.

## 3.2 Distillation

The distillation pipeline is illustrated in Figure 4. It largely follows prior work (Jian & Torresani, 2022; Roy et al., 2022; Samuel et al., 2024a), but uses a strong off-the-shelf teacher and our novel image generation method. As noted in Jian & Torresani (2022), images from the base classes (meta-training split) of a few-shot dataset can be pseudo-labeled by a teacher and used for distillation. We include this idea, leaving us with 3 types of distillation training data: (1) training images from base classes, (2) TINT-generated synthetic images of novel classes, and (3) support images from novel classes. The base and synthetic images are pseudo-labeled using a strong off-the-shelf teacher. See Appendix B.4 for more details.

An option would be to use direct label supervision for the synthetic data, skipping the teacher. However, we expect that generating representative images is a harder problem than correctly classifying given images. For example, if the generator mistakenly produces an image that looks more like another class, the student is supervised with pseudo-labels encoding what the image *actually looks like* according to a good teacher, rather than with labels representing what the *generator believed* it was doing. Thus, a mistake by the generator may lead to a sub-optimal distribution over which the empirical loss is measured, but not to an incorrect learning signal akin to a mislabeled training example.

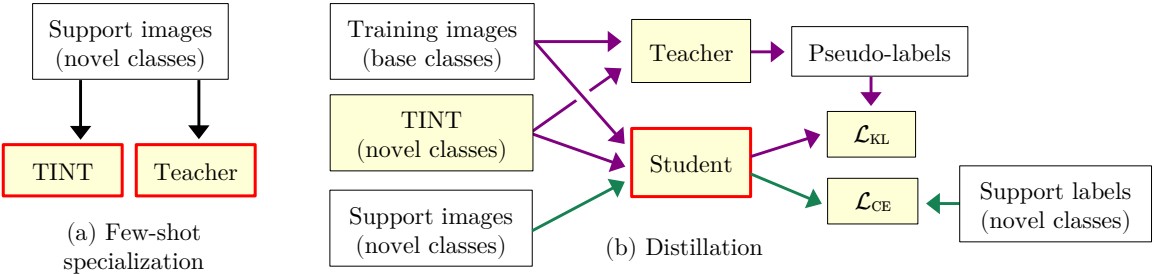

Figure 4: Overview of our few-shot transfer pipeline. First, the TINT generator and teacher are specialized on the novel classes (a), and then a distillation procedure is run to transfer the knowledge from the teacher to the student (b). The modules adjusted in each step are outlined in red.

### 3.3 Efficient multi-episode evaluation

Few-shot classifiers are often evaluated over randomly drawn *episodes*, where each episode contains a random set of novel classes with random support and query examples. For example, results on *miniImageNet* (Vinyals et al., 2016) are typically evaluated over 600 episodes, each containing 5 classes with $K$ (1 or 5) support examples and $Q = 15$ query examples per class. In our pipeline, specialization involves generating thousands of synthetic images, and repeating this process 600 times or more is computationally cumbersome. However, evaluating over a larger query set would use negligible compute in comparison. Instead of the traditional choice of 15 query examples per class, we could easily use the remaining 595 queries per class for miniImageNet. A question then arises: *can we trade episodes for query examples*, i.e. run the evaluation over fewer episodes but with more query examples per episode? How would this affect the statistical significance of the estimate? As a guide for what happens when varying the number of episodes and queries, we present the following theorem:

**Theorem 1** *Let $P, Q \in \mathbb{Z}^+$ be the number of episodes and query examples per episode respectively. Let $a_p \in [0, 1]$ be the true (unknown) accuracy of an evaluated method on episode $p \in \{1, ..., P\}$ and let $a = \mathbb{E}[a_p]$ and $\sigma_a^2 = \mathrm{Var}[a_p]$ be the true (unknown) mean and variance of $a_p$ over i.i.d. episodes $p$.*

*Furthermore, let $\tilde{a}_p$ be our estimate of $a_p$, computed by measuring the empirical accuracy over $Q$ i.i.d. query examples from episode $p$, and let $\tilde{a}$ be the final empirical accuracy computed as the average of $\tilde{a}_p$ over $P$ independently drawn episodes.*

*Then, for any choice of $Q$ and $P$, we have that*

$$\mathbb{E}[\tilde{a}] = a \tag{5}$$

$$\mathrm{Var}[\tilde{a}] = \frac{1}{P} \left( \frac{1}{Q} a(1-a) + \left(1 - \frac{1}{Q}\right) \sigma_a^2 \right). \tag{6}$$

Eq. 5 means that the estimated accuracy $\tilde{a}$ is an unbiased estimate of the true unknown $a$ regardless of $P$ and $Q$. Eq. 6 shows how $P$ and $Q$ affect the estimator variance, and can be used e.g. to determine suitable values for $P$ and $Q$ to reach a certain desired estimator variance. To arrive at Eq. 6, we model the evaluation over each episode as performing $Q$ Bernoulli trials with success rate $a_p$, which makes $\tilde{a}_p$ the mean of $Q$ Bernoulli-distributed variables. We can then derive an expression of $\mathrm{Var}[\tilde{a}]$ using the law of total variance. The full proof is presented in the appendix, Section A.3.

Theorem 1 allows us to study the expected impact of varying $Q$ and $P$. Note that for moderately large $Q$, Eq. 15 can be approximated as

$$\mathrm{Var}[\tilde{a}] = \frac{1}{P} \left( \frac{1}{Q} a(1-a) + \sigma_a^2 \right). \tag{7}$$

As $Q$ increases, $\mathrm{Var}[\tilde{a}]$ approaches $P^{-1}\sigma_a^2$, representing the case where the accuracy of each episode is estimated perfectly, and only the inter-episode variance remains.

Eq. 15 (or Eq. 7) can be used for experiment planning, assessing the expected variance of an evaluation using a given $P$ and $Q$, or giving guidance around the number of evaluation episodes required for demonstrating a certain improvement with a desired statistical significance. As a concrete example, for the LabelHalluc method (Jian & Torresani, 2022), $a \approx 0.87, \sigma \approx 0.05$, giving an estimator variance of 7e-6 over 600 episodes using 25 query examples (5 per class). In preliminary experiments, the accuracy of our method was $a \approx 0.93$ with $\sigma \approx 0.028$. Increasing $P$ to include all remaining examples ($595N$), asserting that we want a similar estimator variance as the original LabelHalluc method and solving for $P$ gives us $P = 118$. We therefore settled for 120 evaluation episodes as a reasonable trade-off between computational requirements and confidence. As can be seen in Table 5, our 95% confidence interval for ResNet12 on miniImageNet is indeed similar to that of the LabelHalluc method (0.50 vs 0.48), showing that the practical results are consistent with the theory.

## 4 Related Work

The state-of-the-art performance on miniImageNet (Vinyals et al., 2016) has increased steadily over the years (Qiao et al., 2018; Oreshkin et al., 2018; Ye et al., 2020; Mangla et al., 2020; Chen et al., 2021; Bateni et al., 2022). Most recent work rely on other techniques than generative distillation, and often use relatively large models. Regarding tiny models, the TRIDENT method (Singh & Jamali-Rad, 2022) produced good results using a model only 4x as large as the Conv4 model (see Section 5.2), but due to their transductive setting, these results are not directly comparable to our work. Nowadays, the top performers are models using extra training data (Hu et al., 2022; Fifty et al., 2024).

Knowledge distillation as a means of improving accuracy of a baseline model or for model compression has been widely studied (Hinton et al., 2015; Gou et al., 2021; Tian et al., 2019; Chen et al., 2022). One notable example is SimCLRv2 (Chen et al., 2020), where a self-supervised model is trained on a large set of unlabeled data, specialized on a small labeled set, and distilled to a potentially smaller model. Like most related methods, the distillation requires a representative unlabeled dataset.

There is a growing interest in using generated training data for supervised learning (Hennicke et al., 2024; Sarıyıldız et al., 2023). Azizi et al. (2023) showed that generated data improves imagenet classification, but did not address the few-shot case or transfer to smaller models. Other works use generated data for few-shot classification, but often in the feature domain and without combining it with distillation (Hariharan & Girshick, 2017; Wang et al., 2018; Xu & Le, 2022). Some prior works combine generative models with distillation. Nguyen et al. (2022) addressed distillation with limited data using a CVAE and MixUp regularization, but do not address the case where the teacher is also a few-shot model, and do not present few-shot classification results on miniImageNet. Ding et al. (2023) uses GANs Goodfellow et al. (2014) to generate distillation data, also without addressing the few-shot case. Saha et al. (2022) concludes that contrastive pre-training is preferable to using GANs. They also include a distillation step in their pipeline, but assume access to a large unlabeled representative set. In DatasetGAN (Zhang et al., 2021c) and related works (Tritrong et al., 2021), GANs are used to generate pixelwise labeled distillation training images. In contrast to our work, they require manual annotation of GAN-generated images (rather than using inversion techniques) and a large set of unlabeled application-specific images.

The original textual inversion method (Gal et al., 2022) is related to a larger body of literature on text-to-image personalization (Gal et al., 2023; Tewel et al., 2023; Cohen et al., 2022; Kumari et al., 2023; Ruiz et al., 2023). Null-text inversion (Mokady et al., 2023) is also related to GAN inversion (Xia et al., 2022). Most works in these directions have image generation or editing as the intended application, and don't show results for few-shot classification.

A recent line of work (Samuel et al., 2024b;a) use diffusion models for long-tail generation, with few-shot learning as one application examples. As for our work, their few-shot pipeline can be derived back to LabelHalluc Jian & Torresani (2022). Their *SeedSelect* method (Samuel et al., 2024b) works by optimizing a *seed* (initial latent noise) to produce images better matching a set of input images. Similarity to target images is measured using direct similarity between the VQ-VAE latents and CLIP embeddings of the generated image and the inputs. With *norm-aware optimization (NAO)*, Samuel et al. (2024a) suggest a way to interpoloate and find centroids of seeds, avoiding areas with unusual norms in the seed/noise space. Taken together, they

Table 1: FID measure (64 features) in comparison with NAO+SeedSelect (Samuel et al., 2024a) and for selected ablations.

| Method | FID ↓ |
|---|---|
| NAO+SS | 13.0 |
| Plain TI | 8.0 |
| TI + superres + limited $t$ range | 4.0 |
| Full TINT | **2.7** |

Table 2: Comparing teacher and synthetic data options (1k or 4k synthetic examples per class) for our method and NAO+SeedSelect (Samuel et al., 2024a).

| Synthetic data | Synthetic teacher | Base teacher | Acc |
|---|---|---|---|
| TINT (4k) | LH | LH | 87.7 |
| TINT (4k) | None | LH | 89.5 |
| TINT (4k) | None | PMF | 91.9 |
| TINT (4k) | PMF | PMF | **94.1** |
| NAO+SS (1k) | None | PMF | 89.3 |
| NAO+SS (1k) | PMF | PMF | 90.4 |
| TINT (1k) | PMF | PMF | **93.6** |

generate novel images resembling the support images by inverting each support example to the noise side of the diffusion model, computing the centroid using NAO, constructing interpolation paths between the centroid and each inverted support example, picking seeds along these paths, and optimizing each seed using SeedSelect. However, those methods require a known text description of the novel classes, and SeedSelect requires backpropagating the loss through the entire diffusion process, which is very demanding in terms of compute and memory.

Finally, Trabucco et al. (2024) studied using diffusion models for data augmentation, but suggested a different method and evaluated on different datasets. They also made an effort to prevent training data to leak into their evaluation. We note that their *data-centric leakage prevention* consists of hiding the known class prompts to the diffusion model, which we also do. To illustrate that their method works outside of the domain that stable diffusion was trained on, they evaluate on a new dataset, *Leafy Spurge*. While the core dataset has been made publicly available (Doherty et al., 2024), their particular setup ($250x250$ regions classified by a domain expert) has not. Therefore, a direct comparison on this dataset is not possible. We do compare to their core image generation method in Section 5.3, but used in our overall pipeline.

## 5 Experiments

### 5.1 Generated data quality

We first study the data quality of examples generated using TINT compared to competing methods and simplified versions of our own method. To be useful for distillation, we would like the data to be both diverse and representative of the true classes. A popular way of evaluating synthetic data in these respects is using the FID measure (Heusel et al., 2017). We note that the original measure involves estimating a $2048 \times 2048$ covariance matrix. This is numerically problematic in our case, since there are only 600 examples per class in miniImageNet. Instead, following Samuel et al. (2024a), we use a feature space of only 64 dimensions, using pooled features from an earlier layer of the Inception V3 model. We generated 300 random images per class, compared to 300 randomly drawn real images of the same class, repeated the process for 5 episodes (with 5 classes per episode) and computed the average FID score. The results are shown in Table 1. Even though this is a rather coarse measure, it shows that the TINT method in some respect generates images following the novel class images distribution better than competing methods.

Our main goal is not to generate data that is pleasing to look at, and we therefore opt against using human assessment as quality metric. Diversity and quality of the generated samples is in our case only important if it helps us improve the final classifier accuracy. For example, if the generated images show very little diversity (e.g. just generating the support examples over and over again with some noise added), this would not help the classifier distillation training much, and the few-shot performance would be poor. In that sense, final classifier accuracy is already an implicit measure of generated data diversity and quality.

Figure 5: More qualitative examples of support (top) and generated (bottom) examples for two classes from miniImageNet. Left: *guitar*, right: *roundworm* (failure example).

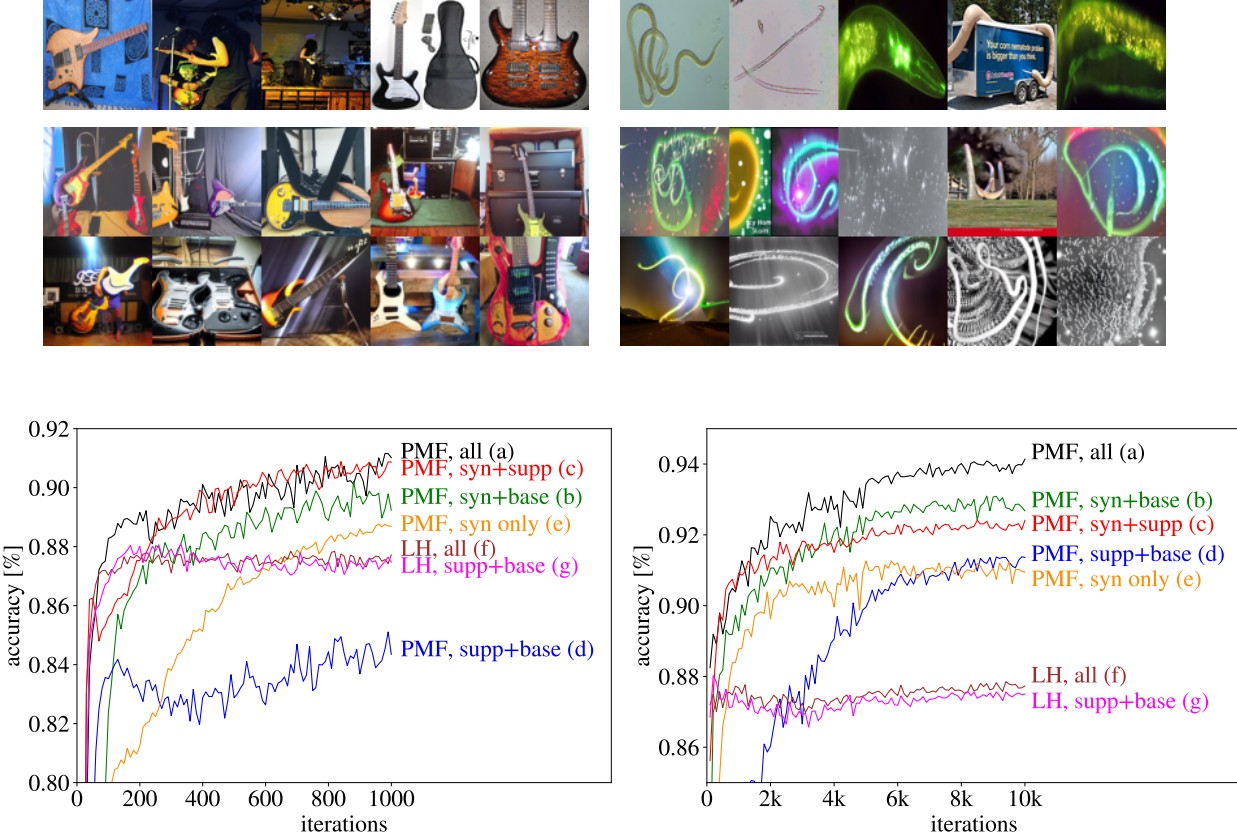

Figure 6: Accuracy over iterations for two choices of teachers and data sources. PMF denotes the P>M>F teacher by Hu et al. (2022) (DINO + ProtoNet), and LH denotes the LabelHalluc teacher (Jian & Torresani, 2022) using ResNet12 and logistic classifier. The left plot is a zoomed-in version of the right plot (note the scale on both axes)

For a better understanding of the method limitations, we include an example of a failure case in Figure 5. Here, the inversion procedure fails to capture the essence of the *roundworm* class from imagenet, and generated images are frequently reminiscent of atmospherical phenomena or pyrotechnics rather than of worms. This is understandable based on the two microscopy images present in the support set. Recall however that the negative effects of non-desired variation in the generated images are mitigated by the use of a teacher.

Handling significant domain shift is out of scope for this work. Nevertheless, we include an example of generated aerial images in Appendix C, to give at least a qualitative indication in this direction.

## 5.2 Student and teacher models

As student, we used two of the smallest models that are widely used in the few-shot literature. The *Conv4* model is a tiny 4-layer CNN with only 113k parameters that was used already in the original ProtoNet (Snell et al., 2017). The *ResNet12* model (8M parameters) is significantly larger, but still one of the smaller models used frequently in prior art (Jian & Torresani, 2022; Xie et al., 2022; Luo et al., 2021; Li et al., 2023; Samuel et al., 2024a). The main choice of teacher was the *P>M>F* method (Hu et al., 2022) due to its simplicity and good performance. Appendix B contains more details.

Table 3: Results using various variations of the synthetic data aspect.

| Method | Acc |
|---|---|
| No synthetic data | 91.4 |
| Plain TI | 92.1 |
| SR + TI + full $t$ range | 93.3 |
| SR + TI + limited $t$ range | 93.5 |
| SR + DA-Fusion | 92.8 |
| Full TINT | **94.1** |
| LAION mining | 92.9 |

Table 4: Selected results for the Conv4 backbone on the 5 fixed episodes, miniImageNet validation split compared to the core ideas used in NAO+SeedSelect (Samuel et al., 2024a) and DA-Fusion (Trabucco et al., 2024)

| Method | Syn teacher | Acc |
|---|---|---|
| Plain TI (4k) | PMF | 70.4 |
| SR + DA-Fusion (4k) | PMF | 77.4 |
| TINT (4k) | PMF | **81.7** |
| NAO+SS (1k) | None | 67.3 |
| NAO+SS (1k) | PMF | 64.0 |
| TINT (1k) | PMF | 80.9 |

### 5.3 Initial experiments

To tune selected hyperparameters, perform ablations and selected comparisons to prior work, we first run experiments with the ResNet12 model over a limited set of of 5 fixed episodes from the validation split of miniImageNet. The same episodes were used in all experiments, ensuring comparable results despite the low number of episodes. Initial hyperparameters were based on LabelHalluc (Jian & Torresani, 2022) with minor adjustment described in Appendix B. For the core TINT method, we found that $\alpha = 1$ works best on few-shot classification benchmarks, and used that option for all presented results.

The accuracy over training iterations for a few combinations of data sources and teachers is shown in Figure 6. While the original LabelHalluc accuracy (plot $g$) saturated after around 200 iterations, the accuracy of our full pipeline (plot $a$) kept improving. We therefore also tried training cycles up to 10k iterations (dropping the learning rate by a factor 0.1 after 5k iterations). Interestingly, including synthetic data but keeping the original LabelHalluc teacher gave no significant accuracy gain (plot $f$). Conversely, switching to a stronger teacher without adding more representative data (plot $d$) did lead to increased accuracy, but with a slow onset. When running for only 300 iterations (as in Jian & Torresani (2022)), one may mistakenly believe that this combination has no potential, whereas it will in fact start to outperform LabelHalluc after around 3k iterations. A plausible explanation is that distillation requires significantly more iterations when transferring knowledge between different architectures. The final accuracy of our method after the full 10k iterations is shown in the top part of Table 2. Note that in order to reach the highest accuracy, both a strong teacher and the synthetic data is required.

Table 3 shows ablations where we kept the few-shot pipeline fixed while simplifying the image generation. This also includes mimicking the DA-Fusion method (Trabucco et al., 2024), using TI and noising each support example to a random time step before denoising. We also run one ablation where the generative model was removed altogether and replaced with real data mined from the LAION dataset (Schuhmann et al., 2022) (original SD training data) by comparing CLIP embeddings of the class names and LAION images. Interestingly, this setup performed significantly worse than our full pipeline even though real images and known class names were used. This shows that the generative model contributes with substantial value to the pipeline. More details are provided in Appendix B.5.

We also compared with data generated by th competing NAO+SeedSelect method (Samuel et al., 2024a). The authors provided source code for their core image generation method, but not for the few-shot experiments, and we were unable to reproduce their results exactly. Instead, we used their image generation method as a drop-in replacement of TINT in our few-shot pipeline. Their previous results were reported using 1k generated examples, so we used the same setting for TINT in the comparison. The results (bottom part of Table 2) show that TINT performs better in this context. Some additional comparisons using the Conv4 backbone are shown in Table 4, using the same 5 fixed miniImageNet episodes. No additional hyperparameter tuning was applied here, so the results can likely be improved for both methods. However,

Table 5: Main results on miniImageNet, CUB and CIFAR-FS, 5-way 5-shot, with 95% confidence intervals. *: Uses textual descriptions of classes and external text-based prior knowledge. **: Results reported by Bertinetto et al. (2018). †: Requires text descriptions of the novel classes. ‡: Uses a pre-trained CLIP backbone and was further trained on ImageNet1k Deng et al. (2009), Fungi Brigit Schroeder (2018), MSCOCO Lin et al. (2014) and WikiArt Saleh & Elgammal (2015).

| Method | Extras | miniImageNet | CUB | CIFAR-FS |
|---|---|---|---|---|
| *Conv4 backbone* | | | | |
| ProtoNet Snell et al. (2017) | - | $68.2 \pm 0.66$ | - | $72.0 \pm 0.6$ ** |
| R2-D2 Bertinetto et al. (2018) | - | $65.4 \pm 0.2$ | - | $77.4 \pm 0.2$ |
| MetaQDA Zhang et al. (2021b) | - | $72.64 \pm 0.62$ | - | $77.33 \pm 0.73$ |
| BL++/IlluAugZhang et al. (2021a) | - | - | $79.74 \pm 0.60$ | - |
| MELR Fei et al. (2020) | - | $72.27 \pm 0.35$ | $85.01 \pm 0.32$ | - |
| FRN+TDM Lee et al. (2022) | - | - | $88.89$ | - |
| KSTNet Li et al. (2023) | Text* | $73.72 \pm 0.63$ | - | - |
| TINT (ours) | DINO, SD | $\mathbf{80.29 \pm 1.14}$ | $\mathbf{89.56 \pm 0.80}$ | $\mathbf{85.43 \pm 1.03}$ |
| *ResNet12 backbone* | | | | |
| RENet Kang et al. (2021) | - | $82.58 \pm 0.30$ | $91.11 \pm 0.24$ | $86.60 \pm 0.32$ |
| MELR Fei et al. (2020) | - | $83.40 \pm 0.28$ | $85.01$ | - |
| FRN+TDM Lee et al. (2022) | - | - | $92.80$ | - |
| LabelHalluc Jian & Torresani (2022) | - | $86.54 \pm 0.48$ | - | $90.5 \pm 0.6$ |
| KSTNet Li et al. (2023) | Text* | $82.61 \pm 0.48$ | - | - |
| DiffAlign Roy et al. (2022) | SD, Text† | $88.63 \pm 0.3$ | - | $91.96 \pm 0.5$ |
| NAO+SS Samuel et al. (2024a) | SD, Text† | $\mathbf{93.21 \pm 0.6}$ | $\mathbf{97.92 \pm 0.2}$ | $\mathbf{94.87 \pm 0.4}$ |
| TINT (ours) | DINO, SD | $\mathbf{93.13 \pm 0.50}$ | $94.57 \pm 0.60$ | $93.18 \pm 0.86$ |
| *ViT/B backbone* | | | | |
| P>M>F Hu et al. (2022) | DINO | $98.4$ | $97.0$ | $92.2$ |
| CAML Fifty et al. (2024) | DINO, Misc‡ | $\mathbf{98.6 \pm 0.0}$ | $\mathbf{97.1}$ | $85.5 \pm 0.1$ |

the results show that TINT beats textual inversion and generation using the DA-Fusion (Trabucco et al., 2024) and NAO+SeedSelect (Samuel et al., 2024a) ideas in this context.

## 5.4 Final evaluation

The final evaluation was made using the test splits of 3 popular few-shot datasets: miniImageNet (Vinyals et al., 2016), CUB (Wah et al., 2011) and CIFAR-FS (Bertinetto et al., 2018). Our main results for both student models are summarized in Table 5. The use of additional data is indicated in column *Extras*, where *SD* indicates Stable Diffusion (Rombach et al., 2022) trained on the LAION dataset (Schuhmann et al., 2022) and *DINO* indicates the pre-trained DINO model (Caron et al., 2021). Note that DINO was trained on ImageNet, which causes some semantic overlap between pre-training and novel classes. However, the DINO training is fully unsupervised, and this issue was discussed extensively by Hu et al. (2022). We also acknowledge that a direct comparison between methods with different use of extra data is not fair, as the setting without extra data is of course more challenging. Nevertheless, we chose to include methods that do not rely on extra data in the comparison, to show how large gains that are possible by using pre-trained models in the pipeline.

Prior top Conv4 results are clustered around 72-74% accuracy, while larger networks routinely reach better performance. This may lead practitioners to believe that model size is the main bottleneck for Conv4, and that larger models are required in order to reach higher performance. Our results show that it is indeed possible to push the performance even for the tiny Conv4 model significantly if extra data is allowed. For the ResNet12 backbone, there are more representative methods to compare with. Our method is on par with NAO+SeedSelect (Samuel et al., 2024a) on miniImageNet, but slightly worse on CUB and CIFAR-FS. Note however that their method requires a text description of the classes, which ours does not. On the other hand,

we require an additional pre-trained model (DINO) that they do not. Also recall using NAO+SeedSelect as a drop-in replacement of TINT in our few-shot pipeline leads to lower accuracy (see Section 5.3). Since we used their code for the image generation, this discrepancy is likely explained by some aspect of their training setup outside of the core image generation, indicating that even better results could be possible by using TINT under similar conditions.

## 5.5 Computational performance

Measured using one NVidia A40 GPU, the total time required for specialization to 5 novel classes from one miniImageNet episode is 6.41h, with 5.59h for generating 20k images (4k per class) using TINT and 0.82h for distillation. In contrast, NAO+SeedSelect (Samuel et al., 2024a) requires 47.5h GPU hours for generating 5k images (1k per class). This equals 1.0s per generated image for our method compared to 34.2s for NAO+SeedSelect. See Appendix A.4 for a detailed breakdown.

# 6 Discussion

**Summary** We proposed a new diffusion inversion method combining the properties of textual and null-text inversion. We also showed how this method can be used in few-shot transfer pipelines with accuracy on par with prior state-of-the-art, but with significantly faster training. Furthermore, we demonstrated that significantly higher accuracy than what has been previously reported is possible even for the tiny Conv4 model when extra data is allowed in the training pipeline.

**Limitations** The TINT method is partly constrained by the diffusion model resolution. While a super-resolution method can be used to handle a significant upscaling, the method can not immediately be applied to generate images of higher resolution than supported by the diffusion model. The few-shot pipeline requires that a pre-trained generative model and teacher with a domain-relevant pre-training is available, which may not be the case in niche applications. Furthermore, the specialization is relatively compute-heavy. This can be limiting for applications where specialization is made frequently, and makes full multi-episode evaluation on few-shot benchmarks cumbersome. However, our method is not worse in this respect than previous methods using diffusion models in few-shot settings, and we expect that 6-7 hours of specialization time is acceptable in many applications.

**Fairness of using extra data** Few-shot classification benchmarks have traditionally been used in a strict setting, without allowing extra data. Nowadays, this strict view if often relaxed, and there are several published few-shot results relying on extra data Hu et al. (2022); Fifty et al. (2024); Samuel et al. (2024a). Other benchmarks, e.g. regular ImageNet classification, have undergone similar transitions, where most top results now rely on extra training data. As discussed in Section 5.4, we find it reasonable to compare results with/without extra data as long as the use of extra data is clearly disclosed. We fully acknowledge that the strict setting is more challenging, but note that many real-world applications have no rule against extra data. We believe that practitioners will find it useful to see how large accuracy gains that are achievable by allowing extra data in the training.

**Outlook** Image-generating diffusion models improve in a rapid pace. In our pipeline, it is straight-forward to replace the diffusion model with an improved one. As such models become even more general, we expect more and more applications to be fully solvable using our method, even when small, highly efficient final models are required and only a tiny set of application-specific data is available.

Generative distillation is not restricted to classification. Similar methods could be used for e.g. segmentation by replacing the teacher with a strong (but slow) few-shot segmentation engine. Furthermore, if other information than *text* is suitable for defining classes in a specific application, the conditional *text* input could be replaced with any other conditioning mechanism. For example, for medical applications, we could use a generative model conditioned on e.g. tabular medical data. The textual inversion would then rather be a *medical data inversion*, but the method formulation will still apply.

We hope that this work has shown the potential of using generative models in few-shot distillation, and that it will inspire more research in this direction for a wider range of label-constrained problems.

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

# A Additional method details

## A.1 Pseudo-code

This section presents pseudo-code for the TINT method. Algorithm 1 shows how to specialize the generator given a set of support examples representing novel classes, while Algorithm 2 shows how to generate an example given a specialized generator.

---

**Algorithm 1** Generator specialization using $K$ support examples of class $n$

---

**procedure** GENSPECIALIZE($\mathbf{x}_{n,1:K}^{(s)}$)
  $\mathbf{z}_{0,n,1:K} \leftarrow$ vqvae_encode($\mathbf{x}_{n,1:K}^{(s)}$)
  $\mathbf{v}_n \leftarrow$ textinv($\mathbf{z}_{0,n,1:K}$)
  **for** $k \in \{1,...,K\}$ **do**
    $(\mathbf{u}_{1:t,n,k}, \mathbf{z}_{T,n,k}) \leftarrow$ nulltextinv($\mathbf{z}_{0,n,k}$)
  **end for**
  **return** $(\mathbf{v}_n, \mathbf{u}_{1:t,n,1:K}, \mathbf{z}_{T,n,1:K})$
**end procedure**

---

**Algorithm 2** Generate an example of class $n$ from a specialized generator

---

**procedure** GENGENERATE($\mathbf{v}_n, \mathbf{u}_{1:t,n,1:K}, \mathbf{z}_{T,n,1:K}$)
  **sample** $k \sim \mathcal{U}(1,K)$
  **sample** $\mathbf{n} \sim \mathcal{N}(0, I)$
  $\mathbf{z}' \leftarrow (1-\alpha)\mathbf{z}_{T,n,k} + \alpha\mathbf{n}$
  $\tilde{\mathbf{z}} \leftarrow \mathbf{z}'\|\mathbf{z}'\|^{-1}\big((1-\alpha)\|\mathbf{z}_{n,T,k}\| + \alpha\|\mathbf{n}\|\big)$
  $\mathbf{z}_0 \leftarrow G(\tilde{\mathbf{z}}\,;\mathbf{v}_n, \mathbf{u}_{1:t,n,k})$
  **return** vqvae_decode($\mathbf{z}_0$)
**end procedure**

---

## A.2 Implementation details

The code required to run all experiments was implemented in PyTorch (Paszke et al., 2019) using the Diffusers library (Face, 2014) with Stable Diffusion 1.5. The overall experiment framework was implemented by us from scratch, but includes code parts from prior work (Mokady et al., 2023; Gal et al., 2022; Hu et al., 2022; Jian & Torresani, 2022; Samuel et al., 2024a). Source code is available at TBD[2].

## A.3 Proof of Theorem 1

Computing the estimated accuracy $\tilde{a}_p$ over an episode $p$ involves classifying $Q$ independent query examples using a fixed classifier. This can be modeled as performing $Q$ Bernoulli trials with success rate $a_p$. $\tilde{a}_p$ is then the mean of $Q$ Bernoulli-distributed variables, which follows a binomial distribution with

$$\mathbb{E}[\tilde{a}_p] = a_p \tag{8}$$

$$\mathrm{Var}[\tilde{a}_p] = \frac{1}{Q}a_p(1 - a_p). \tag{9}$$

Considering that episodes are drawn randomly and that each episode may have a different true accuracy $a_p$, we model each $a_p$ as a random variable with $\mathbb{E}[a_p] = a$ and $\mathrm{Var}[a_p] = \sigma_a^2$. Since episodes are drawn randomly and independently, all $a_p$ are i.i.d.

For the expectation, we simply have $\mathbb{E}[\tilde{a}_p|a_p] = \mathbb{E}[a_p] = a$. To determine $\mathrm{Var}[\tilde{a}_p]$, we need to consider two sources of variance: due to estimating each $\tilde{a}_p$ from a finite query set (intra-episode variance) and due to

---

[2]released with the camera-ready version

Table 6: Compute time required for various parts of the method, evaluated on miniImageNet using one NVidia A40 GPU with 48 Gb VRAM.

| | | 1 class | 5 classes |
|---|---|---|---|
| Algorithm step | 1 image | 4k images | 20k images |
| Textual inversion | 0.30s | 0.33 h | 1.67 h |
| Null-text inversion | 0.05s | 0.05 h | 0.25 h |
| Generate images | 0.66s | 0.73 h | 3.67 h |
| **Complete TINT** | **1.01s** | **1.12h** | **5.59h** |
| Distillation | | | 0.82 h |
| **Complete specialization** | | | **6.41h** |

each episode having a different true accuracy (inter-episode variance) based on the intrinsic difficulty of each episode. Starting at the law of total variance (Eve's law) (Blitzstein & Hwang, 2019), we can write

$$\text{Var}[\tilde{a}_p] = \mathbb{E}\big[\text{Var}[\tilde{a}_p|a_p]\big] + \text{Var}\big[\mathbb{E}[\tilde{a}_p|a_p]\big] = \tag{10}$$

$$= \mathbb{E}\left[\frac{1}{Q}a_p(1-a_p)\right] + \text{Var}[a_p] = \tag{11}$$

$$= \frac{1}{Q}\big(\mathbb{E}[a_p] - \mathbb{E}[a_p^2]\big) + \sigma_a^2. \tag{12}$$

From the well-known variance formula $\text{Var}[x] = \mathbb{E}[x^2] - \mathbb{E}[x]^2$, we get $\mathbb{E}[a_p^2] = \text{Var}[a_p] + \mathbb{E}[a_p]^2 = \sigma_a^2 + a^2$ and arrive at

$$\text{Var}[\tilde{a}_p] = \frac{1}{Q}(a - \sigma_a^2 - a^2) + \sigma_a^2 = \tag{13}$$

$$= \frac{1}{Q}a(1-a) + \left(1 - \frac{1}{Q}\right)\sigma_a^2 \tag{14}$$

Our estimate $\tilde{a}$ is computed by averaging $\tilde{a}_p$ over $P$ independently drawn episodes, leading to a final estimate $\tilde{a}$ with $\mathbb{E}[\tilde{a}] = a$ and

$$\text{Var}[\tilde{a}] = \frac{1}{P}\left(\frac{1}{Q}a(1-a) + \left(1 - \frac{1}{Q}\right)\sigma_a^2\right) \tag{15}$$

which concludes the proof.

### A.4 Computational performance

The compute time required for various parts of the method is listed in Table 6. The total compute required for one episode is 6.41 hours. While this is cumbersome for multi-episode evaluation, it is not a problem in practice for applications where specialization is done rarely. Note that the final models are very fast, due to the use of small student models (not evaluated here, since the student models themselves are not our contributions). The compute times were evaluated using an NVidia A40 with 48Gb VRAM, but the method itself requires no more than 16Gb of VRAM. The per-image numbers for the null-text inversion and textual inversion were computed by dividing the time required for these operations with the number of images generated.

## B Additional experiment details

### B.1 Models and pre-training

**Students** The network used in the original ProtoNet method (Snell et al., 2017) was dubbed *Conv4* in later works (Ye et al., 2020; Afrasiyabi et al., 2020; Gidaris et al., 2019). It is made up of 4 blocks, each

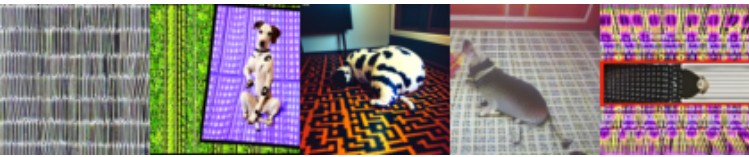

(a) Generated examples using bilinear upsampling

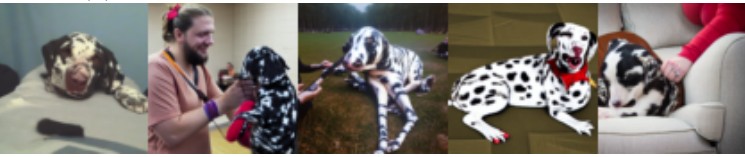

(b) Generated examples using super resolution

Figure 7: Examples of generated images of class *Dalmatian* (test split) with and without super resolution. Examples are hand-picked to illustrate common failure cases.

consisting of a 3×3 convolution, batchnorm, ReLU and 2×2 max pooling. The first block increases the channel count to 64, which is kept throughout the 4 blocks. The *ResNet12* is another popular choice (Jian & Torresani, 2022; Lee et al., 2019; Xie et al., 2022; Ye et al., 2020). With 8M parameters, it is significantly larger than Conv4, but still one of the smaller networks used in prior art. It consists of 4 residual modules, each one consisting of 3 blocks. One block consists of a 3×3 convolution, bachnorm and ReLU. A skip connection connects the input of the first block to the input of the ReLU of the last block. The scale is decreased by a 2×2 max pooling at the end of each module. The first convolution of each module increases the channel count to 64, 128, 256 and 512 channels respectively. Both our implementation and pre-training using IER (Rizve et al., 2021) directly follows Jian & Torresani (2022).

**Teacher**  The teacher backbone used DINO ViT-B/16 with pre-trained weights. The teacher pretraining largely followed the original P>M>F paper (Hu et al., 2022), with some minor variations due to implementation details. We used the ADAM optimizer with initial learning rate 3e-3, lowered by a cosine schedule over 100 epochs. Each epoch consisted of 400 episodes of 5 classes with 5 support examples and 15 query examples per class. The optimizer used a momentum term of 0.9 and weight decay 1e-6. No test-time fine tuning was performed for the teacher, since this is only important when evaluating cross-domain performance.

### B.2   Synthetic data generation

**Diffusion model**  For data generation, we used Stable Diffusion 1.5, with the DDIM scheduler running over 50 time steps. The textual inversion was implemented largely following the original paper (Gal et al., 2022), with 5k optimization iterations, using the ADAM optimizer with fixed learning rate 1e-3. The null-text inversion was run using 10 iterations per timestep, using the ADAM optimizer with a learning rate varying over timesteps according to the original implementation (Mokady et al., 2023).

**Resolution mismatch**  To handle the resolution mismatch between mini-imagenet and the pre-trained Stable Diffusion model, the 84×84 images were first upsampled to 96×96, then upscaled with super resolution using HAT (Chen et al., 2023) to 384×384, and then upsampled again to 512×512. Bilinear interpolation was used in the two upsampling steps. Some hand-picked examples of generated images with and without super-resolution are shown in Figure 7. Note that some images in the top row are completely degenerate (1 and 5), and that there are traces of a grid-like structure also in the other images (carpets in image 2-4), showing why the super resolution step is needed.

### B.3   Datasets

For miniImageNet, we used the common train-val-test split from Ravi & Larochelle (2016). For CUB, we used the common 100/50/50 split (randomly selected) and with images cropped and downsampled to 84x84 resolution. For CIFAR-FS, we used the original splits from Bertinetto et al. (2018). The exact episodes used are provided with the source code. We encourage follow-up work to use the same episodes when comparing

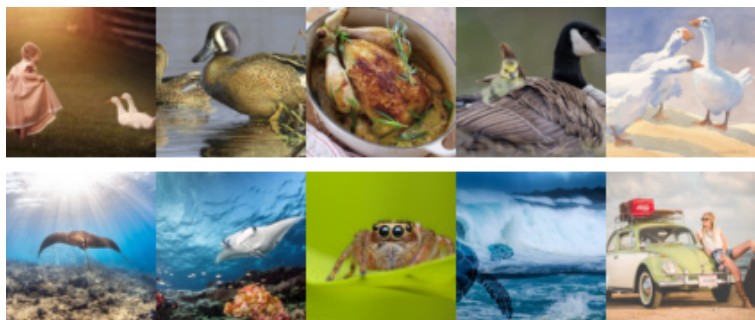

Figure 8: Examples of data mined from LAION (classes *goose* and *rhinoceros beetle* from the mini-imagenet val split). Compare with support examples and generated examples in Figure 2.

with our work, but also to run a final validation on a new set of random episodes to avoid over-engineering hyper-parameters for one specific fixed episode set.

### B.4 Distillation

Similar to prior work (Jian & Torresani, 2022), we used an SGD optimizer with momentum 0.9 and weight decay 5e-4. We adjusted the batch size to 150 to enable constructing a batch from equal parts of 1-3 data sources. The learning rate was set to 0.02 (adjusted slightly due to the change of batch size). For the longer runs, it was reduced by a factor 0.1 after 5k iterations. The loss was computed using cross-entropy for the support examples and KL-divergence for the base and synthetic data sources.

### B.5 LAION data mining details

The LAION dataset consists of 5.85 billion image-text pairs. Working with the entire dataset is unpractical, as it weighs in at 240Tb (and already the metadata requires 800Gb). Furthermore, many images in LAION are of rather poor quality. In Stable Diffusion, the final iterations of the training is performed on subsets of LAION with extra high aesthetic quality, assessed by an automated method. As a practical compromise, we use the subset *improved aesthetics 6+*, consisting of 12M images.

To select images representative of our support classes, we compared the CLIP embeddings of LAION images and of text descriptions of the mini-imagenet validation split classes. We selected the best 1k (at most) matches, while requiring a cosine similarity larger than $T = 0.45$. Some examples of images mined this way are shown in Figure 8. The main issue with training using mined LAION data is that the mined images are often not representative of the actual class (as for the *Beetle* images in Figure 8). Diffusion inversion provides a more consistent way of obtaining images that are visually related to the support examples.

### B.6 Statistical significance

Theorem 1 was used to guide experiment planning, arriving at 120 episodes but evaluating using all remaining query examples per episode as a reasonable trade-off between significance and computation time (see Section A.3). The final 95% confidence intervals reported in Table 5 are computed from the individual episode accuracies using the Students-t distribution as usual, directly following Jian & Torresani (2022).

## C Generated image examples

Figure 9 shows more examples of images generated by TINT. The classes were selected manually, but the support examples and generated images were randomly selected without cherry picking. Note how for example the generated *guitar* images capture the setting of the support examples (focus on electric guitars / stage), rather than producing more generic guitar images. Also note how the generated *king crab* images cover both whole crabs and prepared as food, as well as covering both the red and brown color scales of the

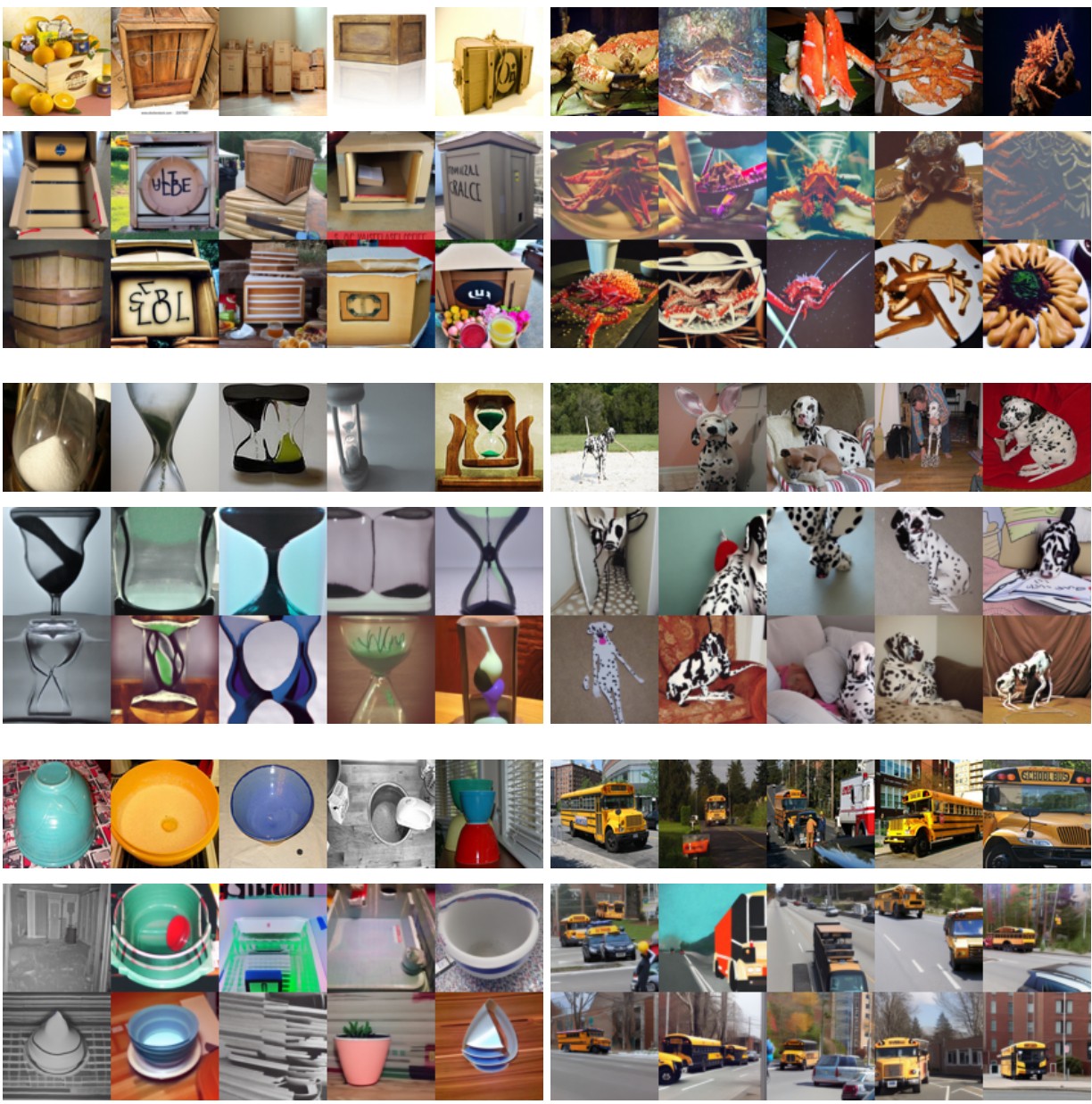

Figure 9: More examples of generated images, classes *crate*, *king crab*, *hourglass*, *dalmation*, *bowl*, and *school bus*. For each class, 5 random support examples are shown on top, followed by 10 randomly selected generated examples.

support examples. Sometimes food-like images are depicted in a brown hue, which is understandable based on the few support examples. The generator also sometimes extrapolates a bit too far in the food direction (lower-right image).

Figure 10 shows a qualitative cross-domain example where the support examples are aerial images from the RSSCN dataset (Cheng et al., 2017). Although we expect that it is possible to improve the image quality by using a dedicated diffusion model, e.g. the model by Khanna et al. (2023), this example shows that the generated images stay reasonably faithful to the support examples even when a general diffusion model is used. As expected, the examples generated with low $\alpha$ values are rather similar to the support examples, while examples generated with higher $\alpha$ present more variation.

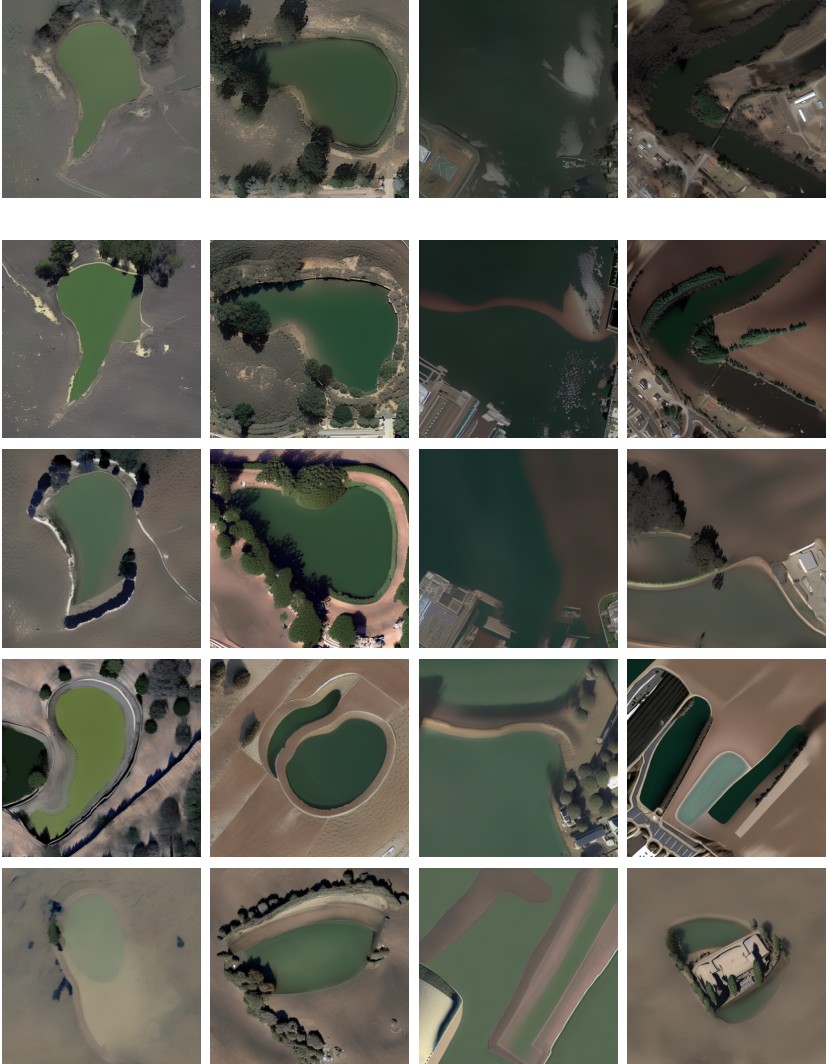

Figure 10: Cross-domain example. Top row: support examples. Remaining rows: generated examples with $\alpha \in \{0.2, 0.4, 0.6, 0.8\}$.

## D    Broader impact

This work has presented how to best train tiny classifiers to maximum accuracy using a minimum of application-specific data. We consider the work foundational in nature. As such, it is not tied to any direct application, but we acknowledge that it may potentially be used for applications with both positive and negative impact, ranging from medical applications to embedded military systems. Regarding the core image generation procedure, TINT, we acknowledge that generative models can be used to spread misinformation through deep fakes. In theory, it may be possible to use TINT to generate deep fakes by applying it to a set of images containing a certain individual, thereby generating similar-looking images of a similar-looking individual in new situations. However, this is a rather crude way of generating deep fakes compared to more direct image editing methods, and we do not expect our method to be adopted for this purpose.

