# OpenReview forum: "Tiny models from tiny data: Textual and null-text inversion for few-shot distillation"
_TMLR — Rejected by TMLR_

### Review · Reviewer_qyZK · 2024-11-30

**Summary Of Contributions:**

This paper introduces a few-shot distillation framework that combines null-text inversion and textural inversion for synthetic data augmentation, leveraging a pre-trained text-to-image diffusion model. By integrating these techniques with a standard teacher-student distillation strategy, the framework demonstrates improved performance on standard few-shot classification benchmarks compared to baseline methods.

**Audience:**

Yes

**Claims And Evidence:**

Yes

**Requested Changes:**

See the weaknesses.

**Strengths And Weaknesses:**

The paper is well-written, easy to follow, and supported by clear and aesthetically pleasing visualizations. Synthetic data generation, particularly leveraging pre-trained foundation models, is a critical research direction, and this paper makes a meaningful contribution by exploring its application in few-shot classification.

I did not identify any major weaknesses in the paper. The method is well-motivated and thoroughly evaluated. However, I have a few open-ended questions that could inspire further exploration or clarification:

1. The authors emphasize the importance of null-text inversion to ensure the generated examples are similar to the samples in the support set. If semantic class labels are known, is null-text inversion still necessary? Could the large-scale object prior inherent in the pre-trained model suffice for this task without additional inversion?

2. Since the framework highlights the importance of "identity-preserving" generation, would it be more straightforward to adopt an identity-preserving image generation approach, such as DreamBooth, as a synthetic data augmentation strategy? How does the proposed method compare to such direct alternatives?

3. The use of a templated prompt like "a photo of #object" reminds me of prompt engineering in zero-shot image classification (e.g., CLIP). Could leveraging a more extensive set of prompts (e.g., those used in CLIP prompt augmentation in https://github.com/openai/CLIP/blob/main/notebooks/Prompt_Engineering_for_ImageNet.ipynb) further enhance the diversity of generated images and, consequently, the framework's performance?

4. While the method is currently focused on image classification, how could it be adapted to dense prediction tasks such as segmentation or depth estimation? Are there inherent challenges in applying the proposed framework to these tasks, given their requirements for spatial consistency?

---

> ### Author Response · Authors · 2024-12-11
>
> We thank the reviewer for interesting open-ended questions, which we will touch upon briefly here.
>
> **Importance of null-text inversion**
>
> In preliminary experiments, we found that using a known text prompt instead of the inversion procedures performs significantly worse. We chose not to include this baseline for compactness, to focus on the case where there is no known textual label. However, in Appendix C, we show qualitatively how e.g. the inverted “guitar” images capture more peculiarities from the dataset than what could be expected from a generic “guitar” prompt only.
>
> **Other “identity-preserving” generation approaches**
>
> DreamBooth and related work in this direction, e.g. using LoRAs [Tewel et al. 2023, Key-locked rank one editing ...] could surely be interesting to examine and is an avenue for future research. DreamBooth and related works often fine-tune the diffusion model itself for specific inputs, which we don’t. There is a certain symmetry and simplicity in only optimizing all conditioning inputs (both conditional and unconditional) in our method that we found appealing. We chose to go for publishing our current results as a baseline at this time, before experimenting with more advanced options.
>
> **Prompt engineering**
>
> We fully agree that more advanced prompt engineering could be useful. In preliminary experiments, we experimented with using the same pool of prompts as the original textual inversion paper did. However, in our context, we didn’t find the difference significant enough, and opted for a single prompt to keep the  pipeline more simple. However, this is indeed a point that could be explored more in future work.
>
> **Other tasks than classification**
>
> This is also an excellent question. We mention in the discussion section (outlook part) that the overall pipeline could be used for e.g. segmentation by replacing the teacher with a strong few-shot segmentation engine. We didn’t elaborate further due to space constraints, but we actually consider this to be one of the more promising future research directions, as we can see this having many real-world applications. Few-shot depth could likely also work, although we don’t see that as useful in applications.
>
> We will focus most of our suggested edits on actual concerns by other reviewers. However, we do suggest adding a brief mention of DreamBooth-like methods and prompt engineering as potential future research.

---

### Review · Reviewer_2NeU · 2024-12-04

**Summary Of Contributions:**

The paper aims to improve the inference latency in the few-shot learning process. To achieve this, it proposes a diffusion inversion method called TINT to generate synthetic data for facilitating knowledge distillation.

**Audience:**

Yes

**Broader Impact Concerns:**

No broader impact concerns.

**Claims And Evidence:**

No

**Requested Changes:**

My suggestion is that the paper would benefit from deeper justification of its components and how they relate to each other. Also, better experiment results should be provided.

**Strengths And Weaknesses:**

Strength:
1. The topic is interesting and relevant, focusing on the use of synthetic data for data augmentation

Weaknesses:
1. The proposed method seems to be a combination of existing techniques, i.e., null-text inversion (NTI), textual inversion (TI), and knowledge distillation (KD). While TMLR does not strictly require high novelty, the performance improvement of TINT over baselines is not substantial—for example, as shown in the ResNet12 backbone results in Table 5.
2. The choice of knowledge distillation in the context of improving inference latency is not well-justified. If inference latency is the main concern, baselines like pruning, quantization, or other model compression techniques should be included for comparison or justification.
3. The theoretical analysis is related to the evaluation of the methods. Its relevance to the rest of the paper is unclear and requires further integration.

All in all, the paper attempts to address an important issue but feels like a collection of loosely related existing techniques that lack strong cohesion.

---

> ### Author Response · Authors · 2024-12-11
>
> We thank the reviewer for valuable comments. We will try to address each concern below.
>
> **Combination of existing techniques**
>
> We agree that this is a combination of already known methods. A lot of research is about finding the right combination of existing techniques. As shown in the ablations, each included technique is required, and we tried to motivate our combination (e.g. discussing the benefits of a teacher in the second part of 3.2, motivating our choice of noise blending etc around Eq 4). We have not seen this combination elsewhere. If we lack a central reference, please provide it and we will happily adjust our claims accordingly.
>
> **Knowledge distillation is not well-justified**
>
> We agree that other techniques than KD (e.g. model pruning and quantization) could be used to improve the inference speed, and we find it to be a good suggestion to cite and relate to them. We note that traditional quantization techniques require data for calibration, which we don’t have (unless using the generated data for this as well, which would be interesting, but it would be a paper of its own). However, there are also data-free approaches, e.g. [Cai et al. 2020, ZeroQ: ...]. Nevertheless, pruning and quantization techniques are partly orthogonal to our method, so their existence does not invalidate the usefulness of our method.
>
> What led us to KD as a design choice was the vast difference in size between SOTA few-shot models, e.g. [Hu et al 2022] using a ViT-B backbone (~86M params), while we wanted to study final models in the Conv4 order of magnitude (113k parameters, i.e. ~0.13% of the large SOTA models). We are not aware of any quantization or model pruning technique that can lead to a reduction of size/inference time of this magnitude. Therefore, we were interested to see if a KD pipeline + synthetic data could be used and study which design choices that worked best within such a setup.
>
> For completeness, we suggest adding references to model pruning and quantization to the related works section. We also suggest mentioning such techniques in the introduction and include our motivation for choosing KD along the lines above.
>
> **Theoretical analysis on estimator variance**
>
> As stated in the first paragraph of Section 3.3, the theoretical analysis of estimator variance is there to support our choice of trading evaluation episodes for query examples. This provides a more convenient and efficient way to evaluate methods involving synthetic data generation on popular few-shot benchmarks. We are not sure what the reviewer specifically finds lacking here. If restructuring this part is critical for raising the paper above the acceptance threshold, we kindly ask for more specific change requests.
>
> **Requested changes**
>
> We hope that adding references and discussion related to quantization and model pruning will serve as the additional justification requested. As for “better experiments”, we are not sure what is expected. A thorough experimental comparison to various quantization/pruning techniques that work in the few-shot setting is an extensive task, which we find out of scope, and we find no other specifically requested experiments.
>
> Since the reviewer answered “No” to the “Claims and evidence” question, we kindly ask the reviewer to specify which of our claims that is not supported by enough evidence, such that we can consider reformulating the claim to better fit what we demonstrate experimentally.

---

### Review · Reviewer_Urqq · 2024-12-05

**Summary Of Contributions:**

- The work introduces TINT, a textual and null-text inversion framework for augmented few-shot distillation. The work uses diffusion models to generate synthetic examples based on the query set and then distills them to a smaller model such as Conv4 and ResNet-12.
- Presents a theoretical analysis to estimate the variance of the accuracy.

**Audience:**

No

**Claims And Evidence:**

Yes

**Requested Changes:**

Please see the weaknesses.

**Strengths And Weaknesses:**

## Strengths

- The paper is well-written for the most part. The experiments include results with confidence intervals, and the authors have provided detailed descriptions of the experimental setup.
- The results are promising. The paper demonstrates that TINT improves over the baseline without synthetic data (Table 3) by almost 3 points in accuracy.
- TINT achieves strong performance when compared to baselines (Table 1).

## Weaknesses
- The related work section is not well-organized. While it includes several relevant papers, the individual paragraphs could be better structured to improve flow. Additionally, the section does not position the current work in relation to these papers. For example, how is the current work different from [a] and [b]? Providing such comparisons would be highly beneficial for the readers.
- The theoretical motivation is weak. A more thorough explanation is needed to justify the necessity of the theoretical argument for expected variance. This aspect is not clearly presented in the paper.
- The paper lacks results on long-tailed datasets. Including experiments with long-tailed datasets would strengthen the work, as few-shot learning is particularly valuable in scenarios with limited labeled data.
- Results with ViT-B are missing. For completeness, it would be helpful if the authors included results with TINT on the ViT-B model in Table 5.

Overall, the paper could be improved by experimenting with long-tailed datasets. Currently, most of the findings are already familiar to the community within the context of general domain datasets.



Nit:
- Page 6, Paragraph 1, “15 query examples …” -> 5 query examples.
- Page 8, Paragraph 2, change 250x250 to $250\times250$.


**References**

[a] Norm-guided latent space exploration for text-to-image generation. NeurIPS 2023.

[b] Effective Data Augmentation With Diffusion Models. ICLR 24.

---

> ### Author Response · Authors · 2024-12-11
>
> We thank the reviewer for valuable comments. We will try to address each concern below.
>
> **Related work / overall structure**
>
> We are unsure in what specific way the related work structure could be improved. Each of the first four paragraphs summarize related work relating to different aspects of our method (prior miniImageNet performance, KD, generated training data, diffusion inversion). The last two paragraphs highlight specific works that we found deserved a more extensive mention. One improvement could be to introduce paragraph headings (as in the Discussion section). We could then also move the two final references, [Samuel et al 2024] and [Trabucco et al 2024], to the “Synthetic training data” paragraph. Would these changes be in line with what the reviewer had in mind for structural improvement?
>
> **Related work / positioning**
>
> We mention the key differences to [Samuel et al 2024] at the end of the 5th paragraph in Related work (Section 4). We include a comparison to [Trabucco et al] in section 5.3, but agree that we could explain the differences better in the related work section. The core method in [Trabucco et al] is doing textual inversion, partially noise an input image (to a timestep t < T) and denoise. This makes the augmented examples retain much of their original overall composition. In contrast, we use noisy latents (at t=T) produced by null-text inversion and blend in noise in the final timestep T. We suggest adding this explanation to the related work section.
>
> **Lacking results on long-tailed datasets**
>
> Long-tailed datasets are constructed to include both classes with a lot of training data and label-constrained classes. In benchmarks, accuracy is generally computed over all classes. If the aim of the paper was to promote TINT as a general-purpose deep augmentation method, and if we had any central claim in this direction, we agree that this would be very suitable to include. However, our research had the specific goal of pushing the performance of tiny models in few-shot settings, and we opted for three of the most common few-shot benchmarks. To keep the paper focused, we opt for not including more experiments at this point.
>
> **Results with ViT-B are missing**
>
> As mentioned above, the paper is focused on pushing the performance of tiny models. This focus is reflected in the title, introduction, choice of overall method (distillation) and experiments. Adding results using a large model like ViT-B would in our opinion make the paper less focused on this goal, and we opt for not including such experiments.
>
> **Most of the findings are already familiar to the community within the context of general domain datasets**
>
> We agree that there is certainly other related work showing the benefit of generative models in label-constrained settings. However, studying the combination of diffusion model inversion and distillation is not as common. Most references using synthetic data from generative models utilize that in a direct way (using the labels used when generating the data), while our use of a teacher can partly mitigate mistakes by the generator, as discussed in the second paragraph in Section 3.2. We have not seen this line of reasoning in other papers.
>
> We listed our contributions in the introduction. We have not seen these contributions anywhere else. If we lack a central reference, please provide it and we will happily adjust our claims accordingly.
>
> **Nit**
>
> “15 query examples …” -> 5 query examples: No, this should in fact be 15 query examples. Common evaluation schemes use 5 support examples per class but evaluate over 15 query examples per class.
>
> 250x250: Thanks, will fix!

---

> > ### Comment · Reviewer_Urqq · 2024-12-12
> > **Response to the authors**
> >
> > Thank you for the response and for clarifying some of the finer points of the paper. Below, I've shared some of my thoughts:
> >
> > **Related Work.**
> > I wanted the structure of the related works to match the 5th paragraph of that section throughout all paragraphs. You can highlight different related work in the field and then point out if you're doing the same or different in comparison to those works. This will help the readers understand the differences between your work and other related work.
> >
> > **Long-tailed and ViT experiments**
> > Thanks for pointing out that the aim of the paper is to focus on smaller models. While I agree that this is the main aim, a reader might be interested to know at what point the utility of the proposed method diminishes. Experiments with long-tailed datasets and ViT will make this work more interesting to the community.

---

> > > ### Author Response · Authors · 2024-12-18
> > >
> > > We greatly appreciate your feedback and suggestions for additional experiments to broaden the paper's appeal.
> > >
> > > __Related work__
> > >
> > > We tried to highlight differences (e.g. last sentences of paragraphs 3-4), but agree that we could elaborate more on this subject. We suggest to edit this section according to your suggestion.
> > >
> > > __Long-tailed and ViT experiments__
> > >
> > > While we agree that further experiments could enhance the paper's scope, we believe that the current work as it stands addresses a significant research problem in the context of leveraging generative models for few-shot learning. In line with the TMLR acceptance criterion, which advises reviewers to assume the criterion is met if there is uncertainty, we respectfully suggest that _some_ individuals in the journal’s audience would indeed find value in our findings, even without the additional experiments you have suggested. We hope this consideration aligns with your understanding of the journal's guidelines.

---

### Author Response · Authors · 2024-12-11
**Overall review response / summary**

We thank all reviewers for their time and for valuable feedback.

In general, we are happy to see that the reviewers mostly found our work well-written and without factual errors. Furthermore, we note that no specific missing references were found invalidating our novelty claims (bullets at the end of the introduction).
We agree that our paper could likely be improved by widening the scope (including more problem sets such as long-tailed recognition, experimental comparisons to other model compression techniques, etc). However, this would also make it less focused on the core motivating problem - pushing the accuracy of tiny classifiers when only a tiny amount of data is available. At this point, we prefer to keep the current focus and not widen the scope. We are, however, committed to improving the clarity and we are prepared to relax our claims if needed.

Summarizing from individual review responses, we suggest the following edits:
- add paragraph headings to the related work section and move the last two references closer to other synthetic data references
- add references and brief discussion related to competing techniques for producing efficient models (pruning and quantization)
- motivate our choice of using KD in contrast to these techniques (see details below)
- add a brief description of [Trabucco et al]
- mention DreamBooth-like methods and prompt engineering as potential future improvements.

One reviewer (2NeU) answered “No” to the “Claims and evidence” question. We are not sure which specific claim that the reviewer found to be problematic. As pointed out in the TMLR acceptance criteria, gaps between claims and evidence can be addressed by authors adjusting (reducing) their claims.  If the current gap is made clear to us, we will be open to rephrasing any claim that the reviewer found problematic.

---

### Decision · Action_Editor_DQUx · 2025-01-26

**Recommendation:** Reject

**Comment:**

After the authors’ revision, two out of the three reviewers recommend rejection. The remaining concerns from the reviewers primarily focus on the contributions and effectiveness of the proposed few-shot distillation strategy.

Reviewers Urqq and 2NeU reiterated that the paper combines existing techniques in a way that, while functional, appears loosely related and lacks strong cohesion. The authors acknowledged that their approach builds on well-known methods, but the reviewers felt that the submission lacked sufficient empirical evaluation and discussion to demonstrate how this combination offers advantages over alternatives or provides meaningful insights. For example, the authors highlighted an interesting point: "Most references using synthetic data from generative models utilize it in a direct way (using the labels used when generating the data), while our use of a teacher can partly mitigate mistakes by the generator, as discussed in the second paragraph of Section 3.2. We have not seen this line of reasoning in other papers." While this observation could be valuable, it has not been studied or evaluated in depth, leaving room for further exploration.

The evaluation of the proposed approach is not comprehensive. The paper primarily focuses on the standard few-shot classification setting and small models. The reviewers suggested evaluating other settings, such as long-tailed datasets, ViT models, and tasks beyond classification. The authors responded by arguing that these additional evaluations are not necessary to support the current claims, a point with which the reviewers did not agree. While I understand the authors' position and acknowledge that these results are beyond the scope of the current claims, I align with the reviewers in suggesting that such analyses would provide valuable insights into the model's behavior – particularly regarding how performance varies with model size and how the method addresses imbalanced distributions, where both few-shot and many-shot classes are present. This would strengthen the paper, especially given that the proposed method combines existing techniques.

Furthermore, the writing of the paper needs improvement in terms of clarity.

Based on these points, I recommend rejection to encourage the authors to further improve their method. They need to convincingly demonstrate the significance of their work and enhance the comprehensiveness of the evaluation.

**Audience:**

Researchers and practitioners working on few-shot learning, image generation, model  efficiency might be interested in reading this paper.

**Claims And Evidence:**

Summary:

This paper proposes TINT, a framework for few-shot distillation that leverages synthetic data generation through a diffusion-based inversion method. The key idea combines textual inversion for diversity and null-text inversion for specificity, enabling the generation of high-quality synthetic examples from a pre-trained text-to-image diffusion model. These examples are used to facilitate knowledge distillation from a large teacher model to a smaller student model, such as Conv4 or ResNet-12, addressing the challenges of inference latency and data scarcity in few-shot learning. TINT demonstrates state-of-the-art performance on standard few-shot classification benchmarks and offers significant computational efficiency. Additionally, the paper provides a theoretical analysis of accuracy estimator variance, optimizing the evaluation process over multiple episodes and query examples. Comparisons with real data further underscore the advantages of generative models in few-shot distillation.


Claims:

The key claims made in the paper are that the proposed TINT method (1) surpasses baseline generation methods in the few-shot distillation setting, by leveraging the diversity of textual inversion, the specificity of null-text inversion, and the teacher model to partially mitigate errors from the generator; (2) proves effective even for tiny models like Conv4; and (3) demonstrates that employing a generative model in few-shot distillation outperforms the direct use of the original real data. In addition, (4) the theoretical analysis of accuracy estimator variance leads to computationally efficient few-shot evaluation.


Evidence:

The evidence provided in the paper mostly supports the claims, but evidence for Claim 1 is a bit weak. As Reviewer qyZK pointed out, the rationale and necessity of certain design choices are unclear, as they are neither empirically validated nor compared through ablations against alternative designs, such as those addressing null-text inversion and identity preservation.

Moreover, Reviewer 2NeU noted that the choice of knowledge distillation for improving inference latency is not well-justified and recommended comparing it with alternative strategies, such as pruning, quantization, or other model compression techniques, for further justification.

The results are not strong enough to support the proposed TINT strategy. While the paper primarily focuses on the standard few-shot classification setting and tiny models like Conv4, Reviewer Urqq suggested that a more comprehensive evaluation, including generalized settings such as long-tail distributions and performance on larger models like ViT, would provide valuable insights into the proposed method's behavior and enhance the overall thoroughness of the evaluation.

For Claim 4 regarding the theoretical analysis, there is general consensus among the reviewers that it is only loosely relevant to the overall paper, with weak motivation. A more thorough explanation and improved presentation are needed.

Claims 2 and 3 are well supported by the experimental results.